# Non-Adversarial Mapping with VAEs

**Yedid Hoshen**
Facebook AI Research

## Abstract

The study of cross-domain mapping without supervision has recently attracted much attention. Much of the recent progress was enabled by the use of adversarial training as well as cycle constraints. The practical difficulty of adversarial training motivates research into non-adversarial methods. In a recent paper, it was shown that cross-domain mapping is possible without the use of cycles or GANs. Although promising, this approach suffers from several drawbacks including costly inference and an optimization variable for every training example preventing the method from using large training sets. We present an alternative approach which is able to achieve non-adversarial mapping using a novel form of Variational Auto-Encoder. Our method is much faster at inference time, is able to leverage large datasets and has a simple interpretation.

## 1   Introduction

Making analogies is a key component in human intelligence. Evidence for that is the significant component of analogy questions in standardized intelligence tests. In order to achieve true artificial intelligence it is likely that computers would need to be able to learn to make analogies. Although making analogies with full supervision is very related to standard supervised learning which has been at the core of machine learning research, the unsupervised setting which is far more challenging, has received less attention. The task has been attempted over the last few decades, but positive results have only recently been achieved for making analogies across very distant domains.

Let us consider two domains $\mathcal{X}$ and $\mathcal{Y}$. Each domain contains an unordered set of samples. In the unsupervised setting, we do not have correspondences between the samples of the two domains. There are two different types of unsupervised analogies that can be learned between different domains: matching and mapping. The matching task sets to find for each sample $x$ in domain $\mathcal{X}$, a sample in domain $\mathcal{Y}$ which is the most analogous to it. The degree of analogy is either defined by humans or a more objective computerized measure. The mapping task attempts to map each sample $x$ in the $\mathcal{X}$ domain to a new sample such that it appears to come from the $\mathcal{Y}$ domain while still being analogous to $x$. A special case is exact matching [1] when the exact analogy for $x$ appears in the $\mathcal{Y}$ domain training set. In this case the optimal match is also the optimal mapped analogy.

In this work we tackle unsupervised mapping across domains. This task has only recently been successfully attempted between sufficiently different domains. Nearly all successful methods but one, have used the combination of two constraints: i) Adversarial domain confusion: the method trains a mapping $T_{XY}(x)$ for mapping between domain $\mathcal{X}$ to domain $\mathcal{Y}$. A discriminator $D_Y()$ is trained to distinguish between mapped $\mathcal{X}$ domain sample $T_{XY}(x)$ and original $\mathcal{Y}$ domain samples. Mapping function $T()$ and discriminator $D()$ are trained in an adversarial fashion iteratively. The objective is that at the end of training, mapped $\mathcal{X}$ domain samples will be indistinguishable from $\mathcal{Y}$ domain samples. ii) The adversarial constraint, on its own, is not able to force mapped images to correspond to a semantically related $\mathcal{Y}$ domain sample. An additional circularity constraint was found necessary where mappings are learned from both $\mathcal{X} \rightarrow \mathcal{Y}$ and $\mathcal{Y} \rightarrow \mathcal{X}$. The constraint is that a sample mapped to the other domain and back to the original domain will be unchanged by this mapping. This was found effective at preserving semantic information across mappings.

Despite the effectiveness of the two constraints, they present several issues. Adversarial optimization is a saddle point problem and its optimization is hard. This makes architecture selection challenging. Scaling to high resolutions is difficult for adversarial training. This has only very recently been achieved, using cascades of generators, using a slow and complex training regime. Combining such high resolution generative architectures with cross-domain mapping presents a formidable challenge. The circularity constraint presents its own challenges: the assumption that the mapping functions $T_{XY}$ and $T_{YX}$ are inverse functions, is equivalent to requiring the transformation between the two domains to be one-to-one. This is not true even for the simplest transformations such as colorization or super-resolution.

Hoshen and Wolf have recently introduced, NAM [2, 3] - a method for unsupervised mapping which does not rely on adversarial training during mapping or on circularity. The main idea behind NAM is to parametrize domain $\mathcal{X}$ by a pre-trained parametric generator $G(z)$, taking parameters $z$ which they call the latent code. The generator is trained using some off-the-shelf method such as unconditional GAN (e.g. DCGAN [4], StackGAN [5], PGGAN [6]) or VAE [7]. NAM also receives a set of training samples from the $\mathcal{Y}$ domain. NAM attempts to learn two sets of variables, i) weights of mapping function $T_{XY}()$ ii) latent code $z_y$ for every training sample $y$. The latent code is optimized such that $y = T(G(z_y))$. Both $T_{XY}()$ and the set of latent codes $\{z_y | y \in \mathcal{Y}\}$ are trained jointly end-to-end.

NAM suffers from several significant issues: i) NAM learns a latent variable $z_y$ per training sample $y$. This means that every image only contributes to a single code during an epoch, which causes an imbalance between the training speeds of the mapping and the codes. ii) NAM does not have a forward encoder, mapping a sample $y$ to its $\mathcal{X}$ domain analogy. Instead it is learned by optimization $\|T(G(z_y)), y\|$. This can require many iterations for a single sample making it quite slow. iii) Although one of the greatest advantages of NAM is the ability to obtain multiple solutions – every solution requires solving the optimization problem in (ii), and there is no guarantee on the variability between the solutions obtained, It simply relies on the non-convexity of neural networks.

In this paper we present a new method VAE-NAM, which overcomes the aforementioned issues with NAM. VAE-NAM replaces the per-example learned latent codes, by a forward encoder $E()$, regressing example $y$ directly to code $z_y$. Learning a simple encoder encounters the same issues as with the circularity constraint. We choose instead to have a stochastic encoder that learns a latent distribution similar to VAE. Along with the the abilities (shared with NAM) to model many-to-one or one-to-many mappings and to enjoy non-adversarial training of the mapping stage, our final formulation has the following advantages: a feedforward mapping for analogies and no reliance on multiple iterations during evaluation, a principled parametric model for the set of possible multiple solutions and a clear analogy to VAEs, allowing direct transfer of VAE research to VAE-NAM.

## 2   Related Work

*Unsupervised domain alignment:* Mapping across similar domains without supervision has been successfully achieved by classical methods such as Congealing [8]. Unsupervised translation across very different domains has only very recently began to generate strong results, due to the advent of generative adversarial networks (GANs), and all state-of-the-art unsupervised translation methods we are aware of employ GAN technology. The most popular additional constraint is cycle-consistency: enforcing that samples mapped to the other domain and back are unchanged. This approach is taken by DiscoGAN [9], CycleGAN [10] and DualGAN [11]. Recently, StarGAN [12] extended the approach to more than two domains. Circularity has recently also been applied by [13] for solving the task of unsupervised word translation. Another type of constraint was proposed by CoGAN [14] and later extended by UNIT [15] - employing a shared latent space. UNIT learns a Variational Auto-encoder (VAE) [16] for each of the domains and adversarially ensures that both share the same latent space. Although CoGAN showed that the shared latent constraint is weaker than the circularity constraint, UNIT showed that the combination yields the best result. Similarly to the code released with UNIT [15], our work uses for some of the tasks a VGG "perceptual loss" that employs an Imagenet pre-trained network. This is a generic feature extraction method that is not domain specific (and is therefore still unsupervised). Our work is built upon NAM, but uses a learned probabilistic encoder rather than learning the codes directly, which has many advantages described in detail in later sections.

*Unconditional Generative Modeling:* Many methods were proposed for generative modeling of image distributions. Currently the most popular approaches rely on GANs and VAEs [7]. GAN-based methods are plagued by instability during training. Many methods were proposed to address this issue for unconditional generation, e.g., [17, 18, 19]. The modifications are typically not employed in cross-domain mapping works. Our method trains a generative model (typically a GAN), in the $\mathcal{X}$ domain completely independently from the $\mathcal{Y}$ domain, and can directly benefit from the latest advancements in the unconditional image generation literature. GLO [20] is an alternative to GAN, which iteratively fits per-image latent vectors (starting from random "noise") and learns a mapping $G()$ between the noise vectors and the training images. GLO is trained using a reconstruction loss, minimizing the difference between the training images and those generated from the noise vectors. Differently from our approach is tackles unconditional generation rather than domain mapping.

## 3 VAE-NAM

In this section we present our proposed method: VAE-NAM. Formally the task of unsupervised domain mapping can be written as follows: Let $\mathcal{X}$ and $\mathcal{Y}$ be two image domains. The task is to find transformation $T()$ such that for every $x \in \mathcal{X}$, the mapped $T(x)$ is the analogy of $x$ in the $\mathcal{Y}$ domain. In this paper we specialize to image domains, however we speculate that our method can be transfered to non-image domains.

### 3.1 Non-Adversarial Mapping (NAM)

NAM [3] is a recent method for unsupervised mapping across image domains. It takes as input an unconditional model of the $\mathcal{X}$ domain, as well as a set of $\mathcal{Y}$ domain training images $\{y\}$. If a set of $\mathcal{X}$ domain training images was given as input, NAM assumes it is used to train a parametric unconditional generative model $G(z)$ by some other method. NAM does not specify requirements on this method as long as the resulting model satisfies: i) compactness: for every allowed set of parameters $z$, the generated image $G(z)$ lies within the $\mathcal{X}$ domain. ii) completeness: for every possible image $x \in \mathcal{X}$, there exists a set of parameters $z_x$ such that $x = G(z_x)$. Unconditional generative modeling is a very active research field, and althought no technique known to us is able to satisfy the requirements exactly (for interesting datasets), some good approximations exist e.g. state-of-the-art GANs or VAEs.

Given a pre-trained $\mathcal{X}$ domain generative model $G(z)$ and a set of $\mathcal{Y}$ domain training images $\{y\}$, NAM estimates two sets of variables: i) Parameters of transformation $T()$ from domain $\mathcal{X}$ to domain $\mathcal{Y}$ ii) A latent code $z_y$ for every $\mathcal{Y}$ domain training image so that the generated $\mathcal{X}$ domain image from this latent code $G(z_y)$ maps to $\mathcal{Y}$ domain image $y$. The entire optimization problem is:

$$argmin_{T,\{z_y\}} \sum_{y \in Y} \|T(G(z_y)), y\| \tag{1}$$

NAM optimizes the parameters of $T()$ as well as all latent codes jointly $\{z_y\}$. Note that a latent code $z_y$ is estimated for every training image $y$.

NAM has significant advantages over other methods: it does not use adversarial training for learning a mapping, the mapping can be one to many and multiple solutions can be obtained for a single input image. NAM is able to use pre-trained generative models i.e. a generative model of the target domain needs to be estimated only once, and can be mapped to many other domains without retraining.

Unfortunately, the NAM framework also has several issues both at training and evaluation time. Having a latent code for every training image means that the number of variables to be estimated scales with the number of images. Having per-image latent code variables means the images do not directly help each other estimate their latent code, but only very indirectly through the mapping function. Additionally, a gradient step is calculated for every latent code only once per epoch whereas the mapping function has a gradient step once per batch. This causes an imbalance in the learning rates as well as sometimes poor estimation for each latent variable. There are also issues at inference: Evaluation by optimization implies that multiple (hundreds) of feedforward and backprop iterations are required. Furthermore multiple solutions are obtained by solving Eq. 1 multiple times, each with a different random initialization, without a principled model for the set of acceptable solutions.

## 3.2 AE-NAM

We first present a simple method, AE-NAM, that overcomes the issues identified in the previous section.

Similarly to NAM, AE-NAM takes as input a pre-trained generator of the $\mathcal{X}$ domain, $G(z)$, and a set of $\mathcal{Y}$ domain training images $\{y\}$. Differently from NAM, VAE-NAM does not directly estimate a latent code for each training image independently. Instead it learns a feedforward encoder that estimates a code $z_y$ given $y$. We will denote the encoder function $E()$.

The simplest solution is to have a deterministic encoder $z_y = E(y)$, encoding the input $\mathcal{Y}$ domain image $y$ into a latent code of domain $\mathcal{X}$. We denote this method AE-NAM. AE-NAM estimates both $E()$ and $T()$ directly:

$$argmin_{T,E} \sum_{y \in Y} \|T(G(E(y))), y\| \tag{2}$$

The intuition for this method, is that we simply autoencode the $\mathcal{Y}$ domain however with the decoder decomposed into two parts: i) an $\mathcal{X}$ domain prior $G()$ and an $\mathcal{X} \rightarrow \mathcal{Y}$ mapping $T()$. The autoencoder forces $\mathcal{X}$ and $\mathcal{Y}$ to share the same latent space. UNIT [15] attempts to ensure shared latent space using adversarial constraints on the $\mathcal{X}$ and $\mathcal{Y}$ domains. Our method escapes this requirement by using the pre-trained generator prior, which fixes the latent space.

This method has the benefit of simplicity, and fast evaluation due to its feedforward nature. AE-NAM however suffers from a significant drawback - it assumes a one-to-one mapping. In fact it has the same issues as the circularity constraint used by other methods. Every $\mathcal{Y}$ domain image $y$, maps into exactly a single $\mathcal{X}$ domain image $\tilde{x} = G(E(y))$. Such mapping will fail in the cases for which the real transformation is one-to-many e.g.: image colorization. A single gray scale image can map into a whole space of RGB images. A key requirement from a mapping function is the ability to learn multiple possible solutions for a single input.

## 3.3 VAE-NAM

To address the variability issue suffered by AE-NAM we present our final approach: VAE-NAM. VAE-NAM encodes every $\mathcal{Y}$ domain image $y$ into a distribution of latent codes $P(z_y|y)$ rather than just a single deterministic transformation. To learn the probabilistic encoding we use the VAE formulation first developed by Kingma and Welling [7].

In a probabilstic autoencoder, the observed $(y)$ and latent $(z)$ variables are modeled as $p(y, z)$. To infer from this model we would need to compute $p(z|y)$. Using Bayes' law to marginalize over $z$, is hard in the general case. Instead, VAEs use the variational approximation to estimate the posterior distribution $p(z|y)$ by a parametric function $q_\lambda(z|y)$. In our case, instead of directly evaluating $p(z|y)$, we approximate it using a neural network. We choose a probability distribution which is simple and parametric, specifically the Gaussian distribution which is simply parametrized by $\mu_y$ and $\sigma_y^2$ for every sample. We thus train a neural network encoder $E()$ regressing $y$ to $\mu_y$ and $\log(\sigma_y^2)$:

$$(\mu_y, \log \sigma_y^2) = E(y) \tag{3}$$

VAEs optimize the KL Divergence between the true and approximate posteriors $KL(q_\lambda(z|y)|p(z|y))$. It can be shown that for an autoencoder this optimization criterion is equivalent to optimizing:

$$L_{VAE} = -E_{q_\lambda(z|y)}[\log p(y|z)] + KL(q_\lambda(z|y)|p(z)) \tag{4}$$

Where we follow traditional VAEs and choose a normal prior for $p(z)$.

VAEs model $p(y|z)$ as a neural network decoder $D()$, mapping latent code $z$ to reconstruct the original image $y = D(z)$. Differently from VAEs however, we impose a strong domain prior on the decoder. The decoder $D()$ is replaced by a combination of the $\mathcal{X}$ domain generative model $G()$ and the $\mathcal{X} \rightarrow \mathcal{Y}$ mapping function $T()$. The decoder is therefore given by $D(z) = T(G(z))$. VAE-NAM is optimized end-to-end using the re-parametrization trick [7], where we simply use samples from the model, and treat them as deterministic variables. The full optimization loss is therefore given as:

$$L_{VAENAM} = \|T(G(\mu_y + \epsilon * \sigma_y)), y\| + KL(q(z|y)|p(z)) \tag{5}$$

A new sample of normally distributed $\epsilon$ is drawn for every example. Under the normal prior the KL divergence is given by:

$$KL(q(z|y)|p(z)) = \frac{1}{2}(1 + \log(\sigma_y^2) - \mu_y^2 - \sigma_y^2) \tag{6}$$

Instead of using the Euclidean pixel loss used in the original VAE formulation, we use the perceptual loss (the theory is invariant to this modification). Optimization is straight forward, $T()$ and $E()$ can use the same learning rates and we are able to use arbitrarily large amounts of training data (whereas only limited training sets can be used for NAM).

## 3.4 Inference and Multiple Solutions

By the end of training, all 3 networks $E()$, $G()$ and $T()$ have been estimated. To evaluate the solution to a new $\mathcal{Y}$ domain image $y$, we first encode:

$$(\mu_y, \log \sigma_y^2) = E(y) \tag{7}$$

For every component in the code, we independently sample a normally distributed value $\epsilon$ (where $\epsilon^i \sim N(0,1)$):

$$z_y^i = \mu^i(y) + \sigma^i(y) * \epsilon^i \tag{8}$$

We then compute the $\mathcal{X}$ domain analogy by evaluating the generator:

$$\tilde{x} = G(z_y) \tag{9}$$

To obtain multiple solutions we simply sample multiple codes, each generation being a solution. Differently from NAM we now possess a parametric model of the solution space. We can therefore sample the solutions with maximal variety such as the ones along different components with maximal variance, or along a line segment in solution space.

Inference requires just a single evaluation of $E()$ and as many generator evaluations as the number of solutions required. This is far more efficient than NAM which requires hundreds of forward and backward steps for each solution.

## 3.5 Implementation Details

In this section we give a detailed description of the procedure used to generate the experiments presented in this paper.

*Unconditional Generative Models:* VAE-NAM takes as input a generative model of the $\mathcal{X}$ domain. We experimented with a variety of generative models and obtained the best performance with a standard DCGAN For the $32 \times 32$ and $64 \times 64$ resolution domains including Shoes-RGB and Handbags-RGB, SVHN, MNIST and Cars. We generally found that it was helpful to choose the latent code dimension to agree with the dataset complexity therefore we used dimension 32 for MNIST and 100 for the rest.

*Encoder:* For the VAE encoder we used a standard modification of the DCGAN discriminator, where the sigmoid was removed from the output layer, and the output dimension was increased to twice the latent code dimension (to model both $\mu$ and $\log(\sigma^2)$).

*Mapping:* We use a CRN [21] mapping function as used by NAM. We modified the architecture to $64 \times 64$ images by removing the appropriate number of layers so that the central residual blocks will operate on the same $32 \times 32$ resolution as in the original architecture. The number of channels used in the MNIST and SVHN experiments is 8, in all other experiments the number is 32.

*Optimization:* The pre-training of the generative models was done using the code supplied by the authors with standard hyperparameter choices. VAE-NAM was optimized end-to-end over $E()$ and $T()$. Differently from NAM, a single learning rate ($3 \times 10^{-3}$) was used across all parameters as all parameters are updated with every training batch. We also use all $\mathcal{Y}$ domain training images available, rather than just 2000. This is due to VAE-NAM's parametric encoder allowing it to benefit from the gradient updates from every training sample.

Table 1: *MNIST→SVHN* Translation quality measured by translated digit classification accuracy (%)

| CycleGAN | DistanceGAN | NAM | AE-NAM | VAE-NAM |
|----------|-------------|------|--------|---------|
| 17.7 | 26.8 | 31.9 | 16.7 | **51.7** |

Table 2: *SVHN→MNIST* Translation quality measured by translated digit classification accuracy (%)

| CycleGAN | DistanceGAN | NAM | AE-NAM | VAE-NAM |
|----------|-------------|------|--------|---------|
| 26.1 | 26.8 | 33.3 | **37.4** | **37.4** |

## 4 Experiments

In this section we evaluate the proposed VAE-NAM against NAM, which is at present the only unsupervised cross-domain mapping method that does not use adversarial training. We evaluate the visual quality of generations, accuracy of analogies and runtime during evaluation.

### 4.1 Quantitative Results

We conduct several quantitative experiments to benchmark the relative performance of VAE-NAM.

*MNIST→SVHN:* MNIST and SVHN are standard benchmarks for evaluating classification algorithms. Several domain mapping algorithms (including DiscoGAN and NAM) evaluated their performance on mapping between the two datasets. The protocal consists of mapping images from MNIST to SVHN (using DiscoGAN, DistanceGAN, NAM, NAM-AE and NAM-VAE) and are then classified by a pre-trained SVHN classification network. This is used to evaluate the percentage of images mapped to the same label in the other domain. The results can be seen in Tab. 1. VAE-NAM significantly outperformed NAM, DiscoGAN and DistanceGAN [22] on the more challenging MNIST to SVHN task (where much information needs to be hallucinated).

*SVHN→MNIST:* This experiment is performed in exactly the same manner as MNIST→SVHN, with the role of the two datasets reversed. As we can see from Tab. 2, AE-NAM and VAE-NAM outperform all other methods. Note that AE-NAM achieves the same results as VAE-NAM (the best result is achieved with the KL-divergence loss term turned off) due to the many-to-one nature of this task.

*Car2Car:* In the car2car task, each domain consists of a set of distinct car models, displayed at an angle $\theta = \{-90, -75, ..., 90\}$ degrees. The task is to map across the domains such that the model is translated to the other domain while the angle is preserved. We evaluate performance by the root median of the squares of the residuals between the mapped and ground truth orientations. This is measured by a pretrained angle regressor with Network in Network architecture, trained on the $\mathcal{X}$ domain training images. VAE-NAM outperformed NAM on this task (and both significantly outperformed DiscoGAN which presented this benchmark). Per-latent variable optimization is more error prone than training an encoder, hence VAE-NAM's better performance. Due to the one-to-one nature of this task, AE-NAM performed similarly to VAE-NAM.

### 4.2 Qualitative Results

We present a qualitative evaluation of our method, both in comparison to other approaches and an investigation into the effects of the different loss components. We evaluated the different methods of the edges2shoes dataset. It consists of $48000$ images of shoes first collected by [23]. Domain $\mathcal{X}$ is the set of shoe RGB images, whereas domain $\mathcal{Y}$ is the set of their edge maps. No correspondences are given to any of the algorithms during training, the dataset therefore forms an unsupervised analogy dataset.

*Comparison of methods:* Several analogies by VAE-NAM and NAM can be observed in Fig. 1. We can see the VAE-NAM visual quality is comparable to NAM on this dataset, where VAE-NAM adheres better to the ground truth. We evaluated the merits of using VAE-NAM vs. the deterministic AE-NAM. It is evident that using the deterministic AE-NAM yields more precise adherence to target image than VAE-NAM. The AE-NAM result however is significantly less realistic than those obtained

Table 3: Car2Car root median residual error (lower is better).

| DiscoGAN | NAM | VAE-NAM |
|:---:|:---:|:---:|
| 13.81 | 1.47 | **1.38** |

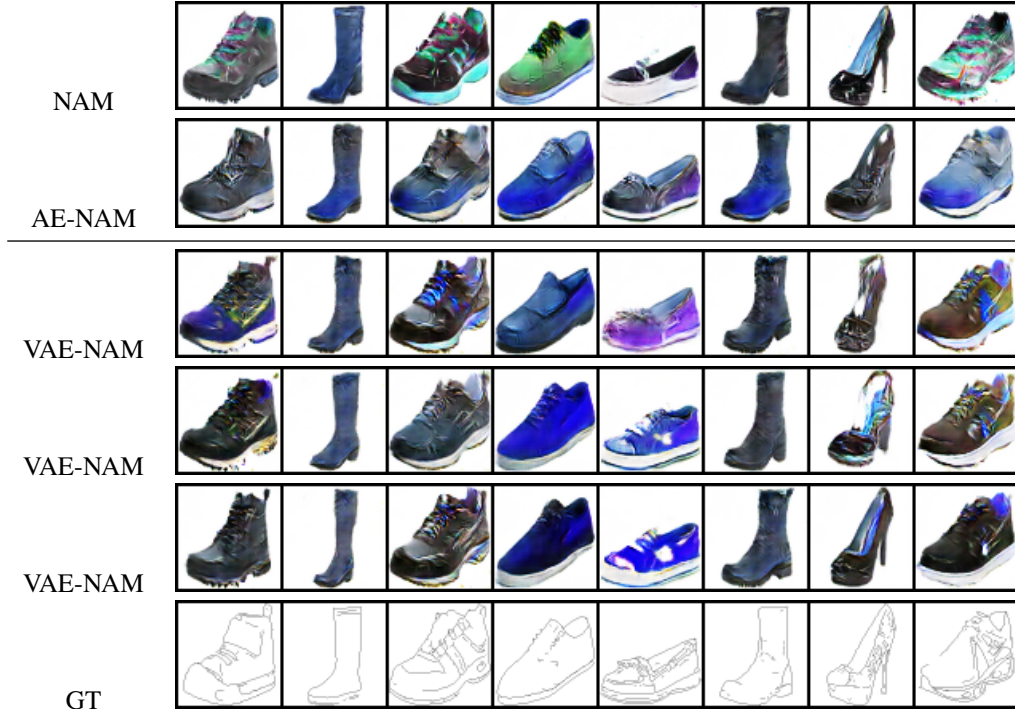

Figure 1: Comparison between NAM (top row) AE-NAM (second row) and VAE-NAM (bottom rows)

from VAE-NAM (e.g. as only a single solution is assigned to each image, it is an average of all possible solutions, and in many cases does not look particularly realistic). AE-NAM also suffers from sample variety issues, it can be seen that analogies of different images have mostly the same color. This is remedied by switching to the VAE-NAM formulation which enables the trade-off between reconstruction accuracy and modeling the variability of the solution space.

*Exploring the latent space parametrized by $\sigma^2(y)$:* VAE-NAM finds a Gaussian probability distribution of $\mathcal{X}$ domain analogies. In Fig. 1 we visually explore the set of analogies encoded by the learned probability distribution. We can see that the learned analogy distribution exhibits much variety which is inherent in the nature of the many-to-one mapping problem.

### 4.3 Runtime Analysis

One of the greatest advantages of VAE-NAM over NAM is the availability of a forward encoder. By the end of training, $T()$ is obtained for NAM, and both $T()$ and $E()$ are obtained for NAM-VAE. At evaluation time, every $y$ domain image requires solving the optimization problem in Eq. 1, this typically requires around 100 iterations each requiring a forward and backward step. For NAM, each iteration requires the evaluation of $G()$ and $T()$ as well as the VGG loss which dominates the evaluation time. For VAE-NAM, only a single evaluation of $E()$ and $G()$ are required. If multiple solutions are required, runtime for both NAM and VAE-NAM increase linearly with the number of solutions.

Evaluating 100 analogies on a P100 GPU and for $64 \times 64$ images took 0.013s for VAE-NAM whereas NAM required 23s. Our proposed method is therefore about $2000\times$ faster than NAM at evaluation

time. We conclude that for applications for which evaluation runtime is important, VAE-NAM should always be preferred over NAM.

## 5 Discussion

Our method VAE-NAM presented a new interpretation for cross-domain mapping. We formulated the mapping as a variational autoencoder on the $\mathcal{Y}$ domain, with special properties. The encoder $E(y)$ maps the $\mathcal{Y}$ domain image into the pre-trained latent space of $\mathcal{X}$. The decoder consists of two parts, the first being a pre-trained $\mathcal{X}$ domain generative model $G(z)$ serving as a transformation prior.

VAE-NAM has some relation to CoGAN and UNIT, which assume that analogous images in the $\mathcal{X}$ and $\mathcal{Y}$ domains share the same latent code. Various constraint are used to ensure this property including adversarial and circularity constraints (although the latter was slightly relaxed in MUNIT). Our model is however significantly different. In CoGAN and later works the only relation between the two domains is through the latent code space $\mathcal{Z}$. In NAM, only the $\mathcal{X}$ domain is directly related to the $\mathcal{Z}$ domain whereas $\mathcal{Y}$ is only directly related to the $\mathcal{X}$ domain (and to $\mathcal{Z}$ only through $\mathcal{X}$). This forces the two domains to share the same latent space without adversarial constraints. VAE-NAM, adds a uni-directional connection between $\mathcal{Y} \to \mathcal{Z}$, however the direct connection between $\mathcal{X} \to \mathcal{Y}$ is maintained. This ensures the two domains share the same latent space while also benefiting from the advantages of having a parametric encoding model.

The success of VAE-NAM suggests that the main component required for unsupervised cross-domain mapping is a strong domain prior. Given such strong prior, the method learns to effectively map between the two domains. The formulation as VAE allows future work to use the vast research developed by the auto-encoder research community for example the Wasserstein autoencoder [24] or recurrence for dealing with temporal data [25].

Generalization to multiple domains occurs naturally, and has a possible cognitive interpretation. Let us assume the agent has a good generative model of a privileged domain, which the agent is very familiar with (in our notation, the domain is denoted $\mathcal{X}$). For every new domain $\mathcal{Y}$, the agent will train a VAE, with the decoder factorized by the known generative model $G_X()$, and a learned mapping function $T_{XY}$. The familiar domain therefore serves as a familiar grounding space, onto which all other knowledge is aligned. Further research is needed to verify if this model in fact accords with human cognitive patterns.

There are multiple ways in which the current work may be extended. Each input sample is mapped so that the latent space is a Gaussian unit ball with identity covariance. This might be too restrictive, and does not fully explore the state of possible multiple analogies. We think that better probabilistic modeling of the latent space can be a fruitful direction.

NAM and VAE-NAM have so far only been applied to image domains. The next step in unsupervised mapping research is to explore the applicability of VAE-NAM to different modalities such as word translation and speech. Non-adversarial mapping is particularly exciting for tasks for which we already have good parametric models. An example of settings for which excellent parametric generators were developed include: face and 3D modeling, as well as general physics simulations and game engines. Successful alignment between the world and a simulation can advance the fields of model based learning and virtual reality.

## 6 Conclusions

We presented VAE-NAM, a method for learning to map across image domains without supervision. Differently from NAM, it uses a parametric stochastic function to map input samples to latent codes, rather than learning the latent code by direct optimization. The model also parameterizes the latent space, and multiple analogies that can arise from a single input sample. VAE-NAM is shown to be better at using large datasets than NAM, is simpler to train, generally converges to better solutions and is much faster at evaluation time.

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
