[Reviews · NeurIPS 2018]

Reviewer 1



The manuscript considers the problem of unsupervised mapping across domains by use of variational auto-encoders (VAE) instead of the current adversarial training (GAN) approach. The approach extends a recently proposed non-adversarial mapping (NAM) framework extracting a mapping from a pretrained generative model to a target domain with the present addition of a probabilistic encoder which maps the target image y to the latent domain z_y, i.e. based on the optimization given in equation 2 of the manuscript. Quality: The issue of mapping between domains is interesting and the framework presented seems new and useful. It however also appears to be a straightforward extension to NAM but providing computational merits and improved performance in terms of the quality of mapping which could warrant publication. The experimentation is somewhat limited and the presentation of the paper could be improved by proofreading the manuscript. Clarity: The basic idea of VAE-NAM appears intuitive, i.e. from a generative model of X formed by G(z) create an encoder that encodes Y in this latent space z_y=E(y) such that a transformation T of the corresponding X resembles Y, i.e. minimize ||T(G(E(y))),y|| as defined in equation 2 treating the encoder with uncertainty using the framework of VAE. However, the manuscript could be clearer in its presentation. First of all the encoder is called C throughout the manuscript and in equation 2 called E which causes confusion. Furthermore, second section of the VAE-NAM description is unclear. Should x be y in the first two lines, i.e. line 167 and 168? In section 4.1 line 234 is referenced to table 1 I believe this should be Figure 1. There is further several spelling issues, i.e., is key component -> is a key component analogies the can -> analogies that can this was found at preserving -> this was found to preserve aforemntioned -> aforementioned direct to code -> directly to code very recently began to generate -> very recently begun to generate Originality: The manuscript appears to be a straightforward extension of NAM to include a variational autoencoder and the work appears somewhat incremental. Significance The results are compelling and the procedure seem to have merits both computationally and in quality of results that may warrant publication. Avoiding the adversarial training and the challenges imposed there also makes the current framework attractive with a potential impact. I thank the authors for their response and find the paper more compelling given their response to the issues raised. I have accordingly increased my evaluation of the paper from just below to marginally above the acceptance threshold.

Reviewer 2



Summary: This paper introduces an AE into NAM to produce the latent code for target image, thus avoids the complex optimization procedure to find the latent code involved in NAM. Then the authors extend the AE-NAM to VAE-NAM, to provide one-to-many ability to the inference procedure. The results reveals the performance of VAE-NAM is much better than NAM and various GAN methods. Comments: 1. Strengths: a) The paper gives quantitative result for NAM-family methods, which make up the shortcomings of the original NAM paper in ICLR workshop 2018; b) The AE trick, although not novel, did solve the problem in original NAM, and the performance is much better in the MNIST->SVHN case. 2. Weakness: a) There is no quantitative comparison between AE-NAM and VAE-NAM. It is necessary to answer that when one-to-many is not concerned, which one, AE-NAM or VAE-NAM should be used. In another word, the superior of VAE-NAM comes from V or AE? b) It contains full of little mistakes or missing references. For example: i. Line 31, and Line 35, mix use 1) and ii); ii. Line 51, 54, 300, 301, missing reference; iii. Equation 2): use E() but the context is discussing C(); iv. Line 174: what is mu_y? missing \ in latex? v. Line 104: gramma error? 3. Overall evaluation: a) I think the quality of this paper is marginally above the acceptance line. But there are too many small errors in the paper

Reviewer 3



The authors address the problem of unsupervised mapping across domains. They summarized two constraints that are usually applied in related work and their drawbacks: 1) Adversarial domain confusion, which is a saddle point problem and is difficult to train and 2) Circularity, which forces the transformation function to become 1-1. They then argue NAM can solve the problem without the above two constraints with some weakness. 1) number of codes depends on number of samples 2) inference by optimization 3) multiple solutions are obtained by multiple restart. The proposed method use an variational autoencoder in place of the directly founding code for each sample, an idea that is easy to understand and makes sense. strength: VAE-NAM is easy to understand and easy to implement. This framework fits any generative model. With this small modification, the 1-1 mapping constraint can be addressed. weakness: The impact is marginal. The major contribution of this paper lies on a tweak of the encoder function, i.e., z_y = C(y), although the final improvement is significant. I do not see the strength from the pictures (in Figure 1) of VAE-NAM in the qualitative research. They just look similar to others. In the the quantitative evaluation, I don't see why AE-NAM is not compared? How does AE-NAM work? The authors mentioned that there is a negative results in SVHN2MNIST line 256, but it is not clear what this refers too. More elaboration is needed. quality: There are some typos. Most annoying is the exchange of C() and E() in the formula. clarity: This paper is reasonably clear originality: This paper is original. The solution makes sense, solves the problems, and is easier to train. significance: Good